# COVID-19 Vaccine Acceptance: A Case Study from Nepal

**Amrit Gaire** [1], **Bimala Panthee** [2,3], **Deepak Basyal** [1], **Atmika Paudel** [2] **and Suresh Panthee** [2,*]

1. Maharajgunj Medical Campus, Institute of Medicine, Tribhuvan University, Kathmandu, Nepal; amritgaire16@gmail.com (A.G.); deepakbasyal2005@gmail.com (D.B.)
2. Sustainable Study and Research Institute, Kathmandu, Nepal; bimupanthee@gmail.com (B.P.); atmikapd@gmail.com (A.P.)
3. School of Nursing and Midwifery, Patan Academy of Health Sciences, Lalitpur, Nepal
* Correspondence: supanthee@gmail.com

**Abstract:** While vaccine acceptance changes over time, and factors determining vaccine acceptance differ depending on disease and region, limited studies have evaluated vaccine acceptance in Nepal. We conducted an online, cross-sectional study to assess COVID-19 vaccine acceptance among Nepalese. Data were collected before and after the vaccine approval in Nepal, during which 576 and 241 responses were obtained, respectively. We found that vaccine acceptance was generally high among Nepalese (93%) and increased after the safety of vaccine was examined by the regulatory bodies (98%). This indicated the role of the national drug regulatory authority to convey drug safety. In addition, we analyzed the predictor(s) of vaccine acceptance. We found that the people who believe that vaccine is an effective measure in preventing and controlling the disease were highly likely to accept vaccination. Given that Nepal had just passed the most devastating wave of COVID-19 during our post-approval data collection, we assume that this might have also played a role in the belief that vaccination is an appropriate approach to combat the pandemic. Likewise, the number of people willing to vaccinate as soon as possible increased from 43% to 86% after approval. Therefore, our results indicate that the government needs to focus on assuring the safety and effectiveness of a vaccine to enhance acceptance. Although fewer responses obtained after vaccine approval might have affected our results, overall, our findings indicate vaccine acceptance is likely to be affected by socio-demographic factors and the attitudes of respondents. This should be carefully considered in the rollout of the vaccination plans in Nepal and countries alike in future.

**Keywords:** COVID-19; vaccine; acceptance; Nepal; regulatory approval

## 1. Introduction

COVID-19 is an infectious disease caused by severe acute respiratory syndrome coronavirus 2 (SARS-CoV-2) [1]. The infection, declared by the World Health Organization to be a global pandemic, can be asymptomatic or range from moderate to severe respiratory sickness, pneumonia, and death [1]. Within a month of the first official global COVID-19 case recorded in China on 31 December 2019, Nepal reported its first case on 23 January 2020 [2–4]. Despite several efforts to combat the spread, the disease is a major problem in Nepal, with several waves of infection and the spread of mutant variants [5]. As of 1 July 2022, more than 6 million deaths have been registered globally, with almost 12,000 in Nepal alone [6]. In the absence of successful treatment options, prevention with disinfection approaches and social distancing is considered the most effective approach to control the spread of infection. However, successful eradication of the disease requires high immunization coverage with an efficient vaccine [7]. Unfortunately, vaccine development is a challenging and time-consuming process, usually taking years to develop. The previous quickest vaccine to be developed, from virus sample to licensure, was four years for mumps in the 1960s [8]. As the availability of any effective vaccine would enormously facilitate our attempts to overcome the disease, the U.S. Food and Drug Administration (FDA) first

aimed to approve a COVID-19 vaccine that would prevent or lessen the severity of illness in at least half of those who received it [9]. As a result, pharmaceutical companies developed many vaccine candidates for SARS-COV-2 in less than a year, and the FDA has now approved a few of them for clinical use. In Nepal, the Department of Drug Administration (DDA) acts as the regulatory body to control the registration of a drug or vaccine candidate. In line with this, on 15 January 2021, the DDA provided authorization for emergency use to "Covishield" ChAdOx1-S-(AZD1222)", a non-replicating viral vector vaccine jointly developed by AstraZeneca and the University of Oxford.

The success of a vaccine relies upon its application and acceptance among the recipients. However, it is typical human behavior to be skeptical of something new, especially when directly related to health. Most of the denial or reluctance in getting vaccinated against COVID-19 arose due to the fear of side effects associated with the vaccine [10]. With time, increased accessibility, and regulatory approval, vaccine acceptance is changing worldwide, including in Nepal. However, no study focusing on the demand and acceptability of the COVID-19 vaccination in Nepal has been conducted. Further, in Nepal, being a middle-income country, the scenario may differ from that of high-income nations [11]. Here, we compared the COVID-19 vaccine acceptance in Nepal before and after the regulatory approval of the vaccine.

## 2. Materials and Methods

### 2.1. Ethics Statement

The protocol of this study was approved by the Institutional Review Board of the Institute of Medicine, Maharajgunj Medical Campus, Nepal (196/(6-11)E$^2$/077/078). Participants' consent was taken before participation in this study.

### 2.2. Study Design, Population, and Sampling

Data collection was performed at two phases: phase 1 (24–30 December 2020) before the vaccine approval by DDA and phase 2 (2–8 August 2021) after the vaccine approval. Due to the limitations in conducting face-to-face research, the questionnaire was hosted on Google Forms and distributed online using various social channels. The general population of Nepal was the intended audience, and samples were drawn from all seven provinces of Nepal (Province 1, Province 2, Bagmati province, Gandaki province, Lumbini province, Karnali province, and Sudhurpacchim province). Participants included in the study were Nepalese ≥18 years old who had access to the internet with an applicable device to fill the questionnaire and could read and understand Nepali/English language. Participants' privacy and identity were protected and kept confidential.

### 2.3. Measures

Data collection used a self-administered questionnaire designed to assess the acceptance of the COVID-19 vaccine. The self-administered questionnaire was designed based on the frameworks of previous studies [12,13]. We conducted a pre-test of the questionnaire among 40 participants who were not included in the research, and we amended it in response to the comments obtained from the pre-test. The framed questionnaire was bilingual (Nepali and English) with the following sections: (1) consent form; (2) socio-demographic characteristics: age, gender, education, marital status, family income, and health condition as well as the perceived risk of COVID-19 infection (perceived risk was defined as the individual feeling of risk associated with COVID-19 infection and was measured on five-point Likert scale and classified as "very less", "less", "fair", "high", and "very high"), and it also covers the impact of the COVID-19 pandemic on the respondents' work, income, and daily life; and (3) acceptance and vaccination preferences for potential COVID-19 vaccinations as well as the influencing factors on vaccination decision-making, vaccine price, its convenience, and respondents' willingness to pay for vaccine and willingness to participate in vaccine trials. Most of the questions were closed-ended and categorical,

and questions such as respondents' health status, pandemic impact, and perceived risk of infection were measured on a five-point Likert scale.

*2.4. Data Analysis and Vaccine Acceptance*

Microsoft Excel 2007 was used for data cleaning and screening, and IBM SPSS statistics version 26 (SPSS Inc., Chicago, IL, USA) was used to analyze all the results. The acceptance of future COVID-19 vaccination was the primary outcome of the study. Participants answering "yes" to "If a new COVID-19 vaccine is successfully developed and approved for listing in the future, would you accept vaccination?" were assigned to the vaccine accept group, and the respondents answering "no" were classified as the vaccine-refusal group. The baseline characteristics of the participants between two phases and between the two groups (vaccine accept vs. vaccine refuse) of phase 1 were compared, and the significance of the association among categorical variables was determined using the chi-square test.

**3. Results**

*3.1. Current Scenario of COVID-19 Cases, Deaths, Vaccination, and Vaccine Approval in Nepal*

With the first case of COVID-19 on 23 January 2020, Nepal has reported a total of more than 0.9 million cases and 11,305 deaths attributed to COVID-19 as of 19 October 2021. Thus far, 28.2% of the Nepalese population has received at least a single shot of vaccine. Nepal experienced a total of three waves of infection (Figure 1), with the third wave being the most severe one. An alarming increase in the case-fatality rate was observed during the third wave [5]. The trend of COVID-19 in Nepal shows that the number of cases and deaths are decreasing, while total vaccinated people are increasing (Figure 1).

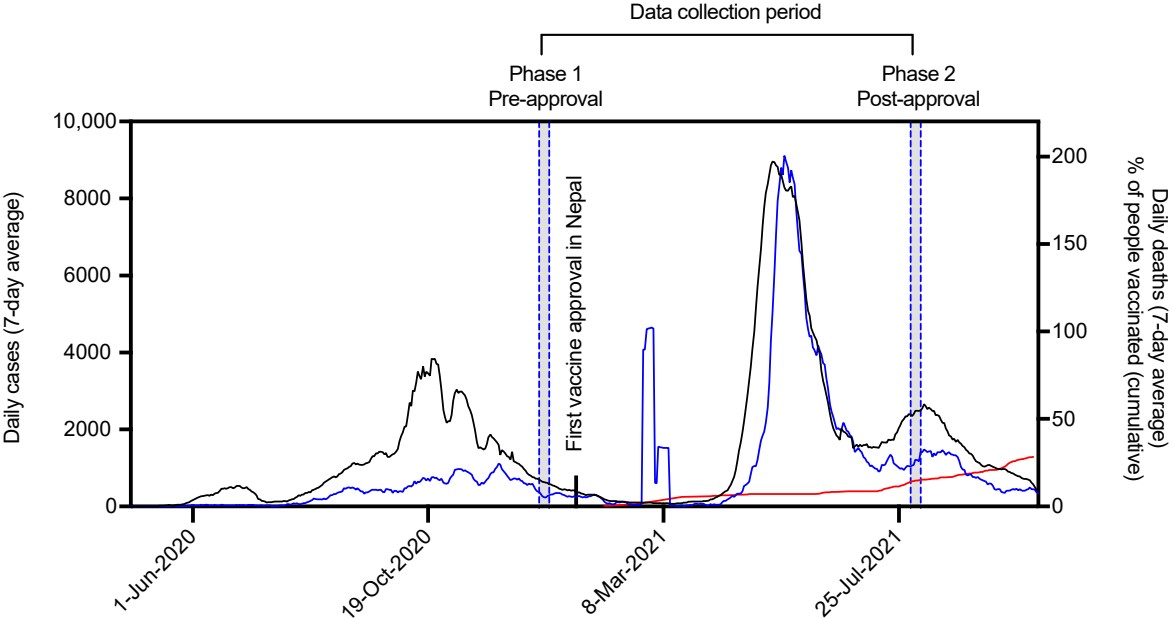

**Figure 1.** Situation analysis of COVID-19 case, deaths, and vaccination in Nepal. Seven-day moving average of the daily COVID-19 infections (black) and deaths (blue). Cumulative percentage of people receiving at least one dose of vaccination is shown by red line. The data were taken from public repository maintained at ourworldindata [14].

The first COVID-19 vaccine was approved in Nepal on January 15. We collected our pre-approval responses when the second wave was at decline. We waited for the third wave, which reached peak in May 2021, to decline before requesting the public's response for post-approval (Figure 1). Within this period of phase 1 and phase 2 data collection, a total of six vaccines were approved in Nepal. With the approval of two additional RNA-

based vaccines on September 2021, a total of eight different vaccines are approved in Nepal (Table 1).

**Table 1.** Vaccines approved in Nepal and their characteristics.

| SN | Vaccine Platform | Type of Candidate Vaccine | Doses and Schedule | Developers | Phase | Approved in Nepal |
|---|---|---|---|---|---|---|
| 1 | Viral vector (Non-replicating) | ChAdOx1-S -(AZD1222) Vaxzevria Covishield | 1–2 Day 0 + 28 | AstraZeneca + University of Oxford | 4 | 15 January 2021 |
| 2 | Inactivated virus | Inactivated SARS-CoV-2 vaccine (Vero cell), vaccine name BBIBP-CorV | 2 Day 0 + 21 | Sinopharm + China National Biotec Group Co + Beijing Institute of Biological Products | 4 | 16 February 2021 |
| 3 | Inactivated virus | Whole-Virion Inactivated SARS-CoV-2 Vaccine (BBV152); Covaxin | 2 Day 0 + 14 | Bharat Biotech International Limited | 3 | 19 March 2021 |
| 4 | Viral vector (Non-replicating) | Gam-COVID-Vac Adeno-based (rAd26-S+rAd5-S) Sputnik V | 2 Day 0 + 21 | Gamaleya Research Institute; Health Ministry of the Russian Federation | 3 | 20 April 2021 |
| 5 | Inactivated virus | CoronaVac; inactivated SARS-CoV-2 vaccine (vero cell) | 2 Day 0 + 14 | Sinovac Research and Development Co., Ltd. | 4 | 4 June 2021 |
| 6 | Viral vector (Non-replicating) | Ad26.COV2.S | 1–2 Day 0 or Day 0 + 56 | Janssen PharmaceuticalJohnson & Johnson | 4 | 30 June 2021 |
| 7 | RNA-based | BNT162b2 (3 LNP-mRNAs) "Comirnaty" | 2 Day 0 + 21 | Pfizer/BioNTech + Fosun Pharma | 4 | 8 September 2021 |
| 8 | RNA-based | mRNA-1273 | 2 Day 0 + 28 | Moderna + National Institute of Allergy and Infectious Diseases (NIAID) | 4 | 15 September 2021 |

*3.2. Study Sample Characteristics and Perceived Impact of COVID-19*

A total of 576 and 241 responses were received during phase 1 and 2 of the study, respectively. In both the phases, younger people (18–30 years of age) accounted for more than two-thirds of the sample. In both the phases, majority were male, unmarried, had completed an education level of undergraduate and above, were working or studying in a healthcare-related sector, and considered that their health status was good. The responses were indifferent in terms of age, gender, religion, race, marital status, perceived health status, and occupation. The respondents in phase 2 differed significantly ($p < 0.05$ by chi-square test) in terms of education, area of residence, and monthly income. Our study population represented that 97% of the respondents were Hindu by religion. Likewise, Brahmin and Chhetri represented 65% of the study population by race. We found that religion or race did not accept vaccine acceptance, corroborating the reports from a previous study [15]. We found that impact of COVID-19 on work and income did not differ significantly during these two phases; however, the number of participants perceiving higher risk of COVID-19 and larger impact on daily life increased during the second phase. The detailed sample characteristics of the population is presented in Table 2.

**Table 2.** Sample population characteristics, their perceived risk, and impact of COVID-19 pre-(*n* = 576) and post- vaccine approval (*n* = 241). Significantly different statistics are indicated in bold.

| Variables | Phase 1 (Pre-Approval) | | Phase 2 (Post-Approval) | | $\chi^2$ | *p* |
|---|---|---|---|---|---|---|
| | Frequency | % | Frequency | % | | |
| **Age** | | | | | | |
| 18–30 | 418 | 72.6 | 171 | 71.0 | | |
| 31–40 | 99 | 17.2 | 48 | 19.9 | 0.97 | 0.61 |
| >40 | 59 | 10.2 | 22 | 9.1 | | |
| **Gender** | | | | | | |
| Female | 274 | 47.6 | 108 | 44.8 | | |
| Male | 302 | 52.4 | 133 | 55.2 | 0.51 | 0.47 |
| **Religion** | | | | | | |
| Hindu | 558 | 96.9 | 235 | 97.5 | 0.24 | 0.40 |
| Others (Muslim, Buddhist, Kirat, Christian) | 18 | 3.1 | 6 | 2.5 | | |
| **Ethnicity** | | | | | | |
| Brahmin/Chhetri | 374 | 64.9 | 156 | 64.7 | 0.12 | 0.93 |
| Madhesi | 102 | 17.7 | 41 | 17.0 | | |
| Janjati and Dalit | 100 | 17.4 | 44 | 18.3 | | |
| **Marital status** | | | | | | |
| Married | 184 | 31.9 | 92 | 38.2 | 2.94 | 0.08 |
| Unmarried | 392 | 68.1 | 149 | 61.8 | | |
| **Highest level of education** | | | | | | |
| Basic and secondary | 61 | 10.6 | 41 | 17 | **10.37** | **0.00** |
| Higher secondary | 160 | 27.8 | 78 | 32.4 | | |
| Undergraduate and above | 355 | 61.6 | 122 | 50.6 | | |
| **Health status** | | | | | | |
| Fair, poor, very poor | 66 | 11.5 | 38 | 15.8 | 2.84 | 0.09 |
| Good, very good | 510 | 88.5 | 203 | 84.2 | | |
| **Main occupation** | | | | | | |
| Agriculture | 64 | 11.1 | 35 | 14.5 | 7.34 | 0.11 |
| Business | 62 | 10.8 | 33 | 13.7 | | |
| Health care students and staff | 239 | 41.5 | 94 | 39 | | |
| Non-medical students | 143 | 24.8 | 44 | 18.3 | | |
| Office worker | 68 | 11.8 | 35 | 14.5 | | |
| **Current area of residence** | | | | | | |
| Province 1 | 68 | 11.8 | 34 | 14.1 | **41.2** | **0.00** |
| Province 2 | 45 | 7.8 | 31 | 12.9 | | |
| Bagmati | 169 | 29.3 | 45 | 18.7 | | |
| Gandaki | 42 | 7.3 | 36 | 14.9 | | |
| Lumbini | 181 | 31.4 | 45 | 18.7 | | |
| Karnali | 38 | 6.6 | 27 | 11.2 | | |
| Sudurpaschim | 33 | 5.7 | 23 | 9.5 | | |
| **Monthly income (NPR)** | | | | | | |
| <10,000 | 65 | 11.3 | 13 | 5.4 | **33.2** | **0.00** |
| 10,000–20,000 | 114 | 19.8 | 27 | 11.2 | | |
| 21,000–30,000 | 151 | 26.2 | 46 | 19.1 | | |
| 31,000–40,000 | 113 | 19.6 | 70 | 29 | | |
| >40,000 | 133 | 23.1 | 85 | 35.3 | | |
| **Perceived risk** | | | | | | |
| Less or very less | 100 | 17.4 | 28 | 11.6 | **17.24** | **0.00** |
| Fair | 320 | 55.6 | 113 | 46.9 | | |
| Large or very large | 156 | 27.1 | 100 | 41.5 | | |
| **Pandemic impact on daily life** | | | | | | |
| Small or very small | 72 | 12.5 | 16 | 6.7 | **7.8** | **0.02** |
| Fair | 345 | 59.9 | 143 | 59.3 | | |
| Large or very large | 159 | 27.6 | 82 | 34 | | |

**Table 2.** *Cont.*

| Variables | Phase 1 (Pre-Approval) | | Phase 2 (Post-Approval) | | $\chi^2$ | $p$ |
|---|---|---|---|---|---|---|
| | Frequency | % | Frequency | % | | |
| **Pandemic impact on Work** | | | | | | |
| Small or very small | 72 | 12.5 | 19 | 7.9 | 4.41 | 0.11 |
| Fair | 255 | 44.3 | 120 | 49.8 | | |
| Large or very large | 249 | 43.2 | 102 | 42.3 | | |
| **Pandemic impact on Income** | | | | | | |
| Small or very small | 94 | 16.3 | 34 | 14.1 | 1.52 | 0.46 |
| Fair | 263 | 45.7 | 121 | 50.2 | | |
| Large or very large | 219 | 38 | 86 | 35.7 | | |
| **Total** | 576 | 100 | 241 | 100 | | |

*3.3. COVID-19 Vaccine Acceptance and Attitudes towards the Vaccine*

A significant majority of the study population agreed to get the COVID-19 vaccine. At phase 1, we found that nearly 94% of the respondents accepted the COVID-19 vaccine, and the acceptance further increased to 98% when the vaccine was made available for free. Among the vaccine-acceptance group, the majority tended to delay the inoculation process to confirm the safety. At phase 2, vaccine acceptance was 98%, and the majority preferred to get the vaccine as soon as possible. Moreover, in phase 1, 58% respondents were willing to pay out of pocket for the vaccine, and this ratio reduced to 36% during phase 2. Among the participants willing to pay, the majority were willing to pay an amount between NPR 100–1999. Although phase 1 results indicated that vaccine price could be an important factor for the decision to get inoculated, the same was not analyzed for phase 2, as the government was already providing vaccines for free to the public.

In general, the respondents perceived that the vaccine is an effective way to combat the pandemic, and vaccine price and convenience are important factors in decision making to inoculate. Although the proportion of respondents agreeing that the COVID-19 vaccine might have side effects increased from 76% in phase 1 to 96% in phase 2, less than one-third of the respondents agreed to participate in a vaccine clinical trial during both phases (Table 3).

**Table 3.** COVID-19 vaccine acceptance and attitude of respondents pre- (*n* = 576) and post- vaccine approval (*n* = 241). Significantly different statistics are indicated in bold.

| Variables | Phase 1 (Pre-Approval) | | Phase 2 (Post-Approval) | | $\chi^2$ | $p$ |
|---|---|---|---|---|---|---|
| | Frequency | % | Frequency | % | | |
| **Vaccine acceptance** | | | | | | |
| Yes | 540 | 93.8 | 237 | 98.3 | **7.69** | **0.00** |
| No | 36 | 6.3 | 4 | 1.7 | | |
| **If yes, when?** | | | | | | |
| As soon as possible | 232 | 43 | 208 | 86.3 | **127.26** | **0.00** |
| Later | 308 | 57 | 33 | 13.7 | | |
| **Vaccination is an effective way** | | | | | | |
| Agree | 506 | 87.8 | 231 | 95.9 | **12.32** | **0.00** |
| Disagree | 70 | 12.2 | 33 | 13.7 | | |
| **Convenience is an important factor** | | | | | | |
| Yes | 544 | 94.4 | 233 | 96.7 | 1.82 | 0.17 |
| No | 32 | 5.6 | 8 | 3.3 | | |
| **Price is an important factor** | | | | | | |
| Agree | 454 | 78.8 | 192 | 79.7 | 0.07 | 0.78 |
| Disagree | 122 | 21.2 | 49 | 20.3 | | |

**Table 3.** *Cont.*

| Variables | Phase 1 (Pre-Approval) | | Phase 2 (Post-Approval) | | $\chi^2$ | *p* |
|---|---|---|---|---|---|---|
| | Frequency | % | Frequency | % | | |
| **COVID-19 vaccine might have mild side effects** | | | | | | |
| Agree | 441 | 76.6 | 231 | 95.9 | **43.30** | **0.00** |
| Disagree | 135 | 23.4 | 10 | 4.1 | | |
| **Participation in vaccine clinical trial** | | | | | | |
| Agree | 184 | 31.9 | 66 | 27.4 | 1.66 | 0.19 |
| Disagree | 392 | 68.1 | 175 | 72.6 | | |
| **Willing to get vaccine if it was free** | | | | | | |
| Willing (probably and definitely) | 565 | 98.1 | - | - | - | - |
| Not sure/ not willing | 11 | 1.9 | | | | |
| **Agree to pay for the vaccine** | | | | | | |
| Yes | 334 | 58 | 88 | 36.5 | **31.37** | **0.00** |
| No | 242 | 42 | 153 | 63.5 | | |
| **If yes, how much (NPR)?** | | | | | | |
| Less than 100 | 30 | 9 | 28 | 31.8 | **31.89** | **0.00** |
| 100–1999 | 237 | 71 | 51 | 58 | | |
| Greater than 2000 | 67 | 20.1 | 9 | 10.2 | | |

### 3.4. Factors Associated with Accepting a COVID-19 Vaccine

Given that 98% of the participants in phase 2 accepted the COVID-19 vaccine, we decided to analyze the respondents from phase 1 to predict the factors associated with vaccine acceptance. Among demographic variables, we found that gender, income, perceived risk, health status, and pandemic effect on daily life and income did not affect the choice to be vaccinated. However, if we take a close look at the significance value, the parameters of perceived risk, pandemic impact on daily life, and pandemic impact on income have a significance value close to 0.05 but not less than 0.05 to designate as a predictor. Based on this, we assume that although these factors might play a role, further studies should evaluate these parameters further. Respondents with young age, unmarried marital status, higher level of education, healthcare workers, people living in Province 1, and people with a larger effect of the pandemic on work were highly likely to accept COVID-19 vaccination. Almost all people living in Provinces 1 and 2 and Bagmati province accepted vaccination, but people from Karnali province had low willingness to vaccination. We found that impact of COVID-19 on work was significantly (chi-square = 22.64, *p* = 0.03) associated with the willingness for vaccination among people, with the evidence that people living in Province 1 and 2 experienced a large or very large impact on their work (47 to 64%), and people living in Karnali province and Sudurpaschim experienced minimal impact on their work. This may be because these provinces were reported low-risk zones, and the number of cases (active cases and new cases) and spreading rate was very low at the time of data collection. It suggests that if people are affected in the area of their work during such a pandemic situation, they accept the vaccination even if it is during the trial phase. We also found that people who believed that COVID-19 vaccination is an effective way to prevent and control the pandemic and who were willing to participate in vaccine clinical trials and pay for the vaccination were more likely to accept the vaccination. Interestingly, people who thought that vaccine convenience is not an important factor and disagreed with the statement that the COVID-19 vaccine might have mild side effects were less willing to accept the vaccination (Table 4).

**Table 4.** Factors associated with COVID-19 vaccination acceptance pre-vaccine approval (*n* = 576). Significant differences are indicated in bold.

| Variables | Vaccine Acceptance | | $\chi^2$ | *p* |
|---|---|---|---|---|
| | Yes n(%) | No n(%) | | |
| **Age** | | | | |
| 18–30 | 400(95.7) | 18(4.3) | | |
| 31–40 | 85 (85.9) | 14 (14.1) | **13.24** | **0.00** |
| >40 | 55 (93.2) | 4 (6.8) | | |
| **Gender** | | | | |
| Female | 253 (92.3) | 21 (7.7) | 1.78 | 0.18 |
| Male | 287 (95) | 15 (5) | | |
| **Marital status** | | | | |
| Married | 161 (87.5) | 23 (12.5) | **18.02** | **0.00** |
| Unmarried | 379 (96.7) | 13 (3.3) | | |
| **Highest level of education** | | | | |
| Basic and secondary | 50 (82) | 11 (18) | **29.35** | **0.00** |
| Higher secondary | 143 (89.4) | 17 (10.6) | | |
| Undergraduate and above | 347 (97.7) | 8 (2.3) | | |
| **Health status** | | | | |
| Fair, poor, very poor | 63 (95.5) | 3 (4.5) | 0.40 | 0.52 |
| Good, very good | 477 (93.5) | 33 (6.5) | | |
| **Main occupation** | | | | |
| Agriculture | 60 (93.8) | 4 (6.2) | **25.5** | **0.00** |
| Business | 50 (80.6) | 12 (19.4) | | |
| Health care students and staff | 234 (97.9) | 5 (2.1) | | |
| Non-medical students | 133 (93) | 10 (7) | | |
| Office worker | 63 (92.6) | 5 (7.4) | | |
| **Current area of residence** | | | | |
| Province 1 | 68(100) | 0(0) | **76.66** | **0.00** |
| Province 2 | 44 (97.8) | 1 (2.2) | | |
| Bagmati | 166 (98.2) | 3 (1.8) | | |
| Gandaki | 38 (90.5) | 4 (9.5) | | |
| Lumbini | 173 (95.6) | 8 (4.4) | | |
| Karnali | 25 (65.8) | 13 (34.2) | | |
| Sudurpaschim | 26 (78.8) | 7 (21.2) | | |
| **Monthly income (NPR)** | | | | |
| <10,000 | 59 (90.8) | 6 (9.2) | 3.32 | 0.50 |
| 10,000–20,000 | 104 (91.2) | 10 (8.8) | | |
| 21,000–30,000 | 143 (94.7) | 8 (5.3) | | |
| 31,000–40,000 | 108 (95.6) | 5 (4.4) | | |
| >40,000 | 126 (94.7) | 7 (5.3) | | |
| **Perceived risk** | | | | |
| Less or very less | 89 (89) | 11 (11) | 4.66 | 0.09 |
| Fair | 303 (94.7) | 17 (5.3) | | |
| Large or very large | 148 (94.9) | 8 (5.1) | | |
| **Pandemic impact on daily life** | | | | |
| Small or very small | 64 (88.9) | 8 (11.1) | 4.89 | 0.08 |
| Fair | 329 (57.1) | 16 (4.6) | | |
| Large or very large | 147 (92.5) | 12 (7.5) | | |
| **Pandemic impact on Work** | | | | |
| Small or very small | 61 (84.7) | 11 (15.3) | **11.50** | **0.00** |
| Fair | 243 (95.3) | 12 (4.7) | | |
| Large or very large | 236 (94.8) | 13 (5.2) | | |
| **Pandemic impact on Income** | | | | |
| Small or very small | 88 (93.6) | 6 (6.4) | 5.79 | 0.059 |
| Fair | 253 (96.2) | 10 (3.8) | | |
| Large or very large | 199 (90.9) | 20 (9.1) | | |

**Table 4.** *Cont.*

| Variables | Vaccine Acceptance | | $\chi^2$ | *p* |
|---|---|---|---|---|
| | Yes n(%) | No n(%) | | |
| **COVID-19 vaccination is an effective way to prevent and control COVID-19** | | | | |
| Agree | 483 (95.5) | 23 (4.5) | **20.64** | **0.00** |
| Disagree | 57 (81.4) | 13 (18.6) | | |
| **Vaccine Convenience is an important factor** | | | | |
| Yes | 518 (95.2) | 26 (4.8) | **36.14** | **0.00** |
| No | 22 (68.8) | 10 (31.3) | | |
| **The COVID-19 vaccine might have side effects such as fever or soreness in the arm** | | | | |
| Agree | 419 (95) | 22 (5) | **5.10** | **0.02** |
| Disagree | 121 (89.6) | 14 (10.4) | | |
| **I would be willing to participate in a clinical trial for the coronavirus (COVID-19) vaccine** | | | | |
| Agree | 181 (98.4) | 3 (1.6) | **9.84** | **0.00** |
| Disagree | 359 (91.6) | 33 (8.4) | | |
| **Do you agree to pay for the COVID-19 vaccine?** | | | | |
| Agree | 327 (97.9) | 7 (2.1) | **23.41** | **0.00** |
| Disagree | 213 (88.0) | 29 (12) | | |
| **If yes, how much (NPR)?** | | | | |
| Less than 100 | 23 (76.7) | 7 (23.3) | **72.45** | **0.00** |
| 100–1999 | 237 (100) | 0 (0) | | |
| Greater than 2000 | 67 (100) | 0 (0) | | |
| **Total** | **540(93.8)** | **36(6.2)** | | |

To confirm if any of these predicted factors were responsible for the increase in vaccine acceptance during phase 2, we first investigated if any of the factors predicted based on phase 1 had a difference in population distribution between the two phases. We found that respondents in two phases were indistinguishable in terms of six predicted factors- age, marital status, occupation, pandemic's impact on work, feeling that vaccine convenience is an important factor, and willingness to participate in a clinical trial. Based on this, we speculated that these were not the reasons behind the increase in vaccine acceptance. The remaining predicted factors had a significant difference between our respondents among the two phases. Thus, we focused on these to look into the role of individual factors towards vaccine acceptance. We found that the respondents on phase 2 were different from phase 1 in terms of education level, area of residence, perception regarding the side effects of COVID-19 vaccine, and willingness to pay. However, the changes related to these were disproportionate regarding vaccine acceptance and thus could not be regarded as responsible factor(s). Interestingly, we found that in phase 2, there was a significant increase in the population who agreed that "COVID-19 vaccination is an effective way to prevent and control COVID-19". Given that a positive attitude regarding this statement was an indicator of vaccine acceptance, we speculate this to be a responsible factor for the increase in vaccine acceptance. The change in respondents' perception of the vaccine in the phase 2 compared to the phase 1 resulted in higher acceptance in the phase 2.

## 4. Discussion

Vaccines are an effective way to prevent contraction and/or reduction of the severity of the diseases. However, the acceptance of getting vaccinated greatly impacts vaccination coverage over an area and the overall pandemic projection. Therefore, a higher vaccine acceptance is expected to help combat the spread of infectious diseases. COVID-19 vaccine acceptance is different depending on countries and has been found to be about 63% in Palestine [16] and 65% in Japan [17]. In the United States, it was found to be about 70%

among the general population [17]. However, underrepresented groups with HIV tended to be much lower acceptors of the COVID-19 vaccination [18]. In countries such as Australia, China, Norway, and India, more than 95% of vaccine acceptance has been reported [17]. Overall, low- and middle-income countries, including Nepal, had higher vaccine acceptance [10], and Nepal had the highest COVID-19 vaccine acceptance among the South-Asian countries studied [19]. However, no study has compared vaccine acceptance before and after the availability or regulatory approval of the vaccine in Nepal.

This study compared COVID-19 vaccine acceptance before and after the availability of vaccines in Nepal. Overall, the COVID-19 vaccine acceptance was high even before its availability although respondents did not want to get vaccinated as soon as the vaccine was available. This was obvious due to safety concerns and fears of unwanted side effects [20,21]. However, COVID-19 vaccine acceptance increased in phase 2 of our research. This could be attributed to several factors, such as vaccine safety and need. First, due to the vaccine's official approval in Nepal and the increasing number of people getting vaccinated, people were convinced of its safety. Next, due to the severity of COVID-19 during the third wave in Nepal and the younger population being the most-affected group [5], the fear of catching COVID-19 surged in Nepal.

We found that married people were less likely to accept vaccination. We assume that married people have a relatively higher feeling of responsibility toward family members [22] and are already taking as little risk as possible, such as minimizing going out unless it is necessary, maintaining social distance, and sanitization. Since they think that they are already well-prepared and are less likely to get infected due to preventive measures other than vaccination, and they are more concerned about vaccine safety and the impact on families if the vaccine is proven to be unsafe and has more side effects than benefits, they are less willing to get vaccinated.

Although we did not evaluate perception towards long-term vaccine safety, it is understandable that people would accept effective and safe vaccines [23–25]. Related to the side effects of the COVID-19 vaccine, people agreeing that COVID-19 vaccination might have mild side effects increased in the second phase, indicating that the respondents were well-informed of the mild side effects experienced by about 40% of the COVID-19 vaccine recipients [26]. Despite mild side effects, vaccinated people were encouraging others to get vaccinated and believed that COVID-19 vaccines effectively control the disease and are safe in the long term [26].

The findings of this study highlighted the impact of pandemic severity and vaccine safety towards vaccine acceptance. Furthermore, the analysis of vaccine acceptance before and after vaccine availability in our research led to the identification of a responsible factor to be the understanding/feeling of people that vaccination is an effective way to prevent and control COVID-19, which might have been different if only one among the two phases was analyzed.

Although we collected the responses from all provinces of Nepal, the online collection of data during a short period of time of a week limited our access to people who had internet service available and could access the rapidly changing global information regarding COVID-19, which might have positively impacted vaccine acceptance. Besides, the higher vaccine acceptance could be due to a sharp increase in COVID-19 cases and deaths during the data collection period. Based on this, the generalization of this data should be performed with caution. Moreover, we cannot deny the bias that might have resulted due to self-administration of the responses.

## 5. Conclusions

This study reflected a high level of acceptance of COVID-19 vaccination among the adult population in Nepal during the pandemic period. This acceptance further increased after the regulatory approval. The public's concerns about vaccine safety may hinder the promotion of vaccine uptake in the future. To expand vaccine uptake in response to the COVID-19 pandemic, immunization programs should be designed to remove barriers

in vaccine price and vaccination convenience. Furthermore, health departments and regulatory bodies should consider regular vaccination safety and education programs for pandemic infectious diseases to improve overall vaccine confidence and the compliance of the public in response to future possible pandemics.

**Author Contributions:** Conceptualization, A.G. and D.B.; methodology, A.G. and B.P.; software, B.P.; validation, B.P., A.P. and S.P.; formal analysis, B.P.; investigation, A.G.; resources, D.B.; data curation, S.P.; writing—original draft preparation, A.G.; writing—review and editing, A.P., B.P. and S.P; visualization, S.P.; supervision, D.B. and S.P.; project administration, S.P. All authors have read and agreed to the published version of the manuscript.

**Funding:** This research received no external funding.

**Institutional Review Board Statement:** The protocol of this study was approved by the Institutional Review Board of the Institute of Medicine, Maharajgunj Medical Campus, Nepal (196/(6-11)E$^2$/077/078). Participants' consent was taken before participation in this study.

**Informed Consent Statement:** Informed consent was obtained from all subjects involved in the study, prior to data collection.

**Data Availability Statement:** The data supporting the findings of this study is available from the corresponding author upon reasonable request.

**Acknowledgments:** The authors thank all the participants who responded to our questionnaire.

**Conflicts of Interest:** The authors declare no conflict of interest.

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
