# Peer review of "COVID-19 Vaccine Acceptance: A Case Study from Nepal"

_covid, doi:10.3390/covid2080075_

Round 1

Reviewer 1 Report

Comment to authors

Abstract

-          Please give more results in this part, in addition, the authors can reduce the first background sentence in the abstract. Only focus on aim.

-          Please provide a summary of the methods including data collection.

-          Please make a logical conclusion based on your results and make a suggestion. The conclusion given in the text is just a pre-determined fact and is very general.

Introduction

-          Please update this: “As of October 19, 2021, nearly 5 million deaths have been registered globally, with over 11000 in Nepal alone[6]”

Materials and Methods

Measures

-          Please give more details about “the perceived risk of COVID-19 infection” in this part.

-          How the authors could check the reality of the answers, as a self-administered questionnaire was used. Please give some information in this part.

Results

-          Generally, A high percentage of people wanted to get the vaccine, even before government evaluation and approval by DDA. This result is interesting, but it may be influenced by the wrong choice of the target community. (Maybe because of the online questionnaire). The choice of the community for research may not have been a true indicator of the whole of Nepal. How did the authors review and confirm this?

-          One of the most influential factors in vaccination is the religious beliefs of popular groups, as well as, races. These can also affect people's trust in the government and drug approval organizations. If possible, these two items should also be investigated or the reason for not investigating should be stated.

Discussion

-          One result was “…perceived risk, health status, and pandemic effect on daily life and income did not affect the choice to be vaccinated.” But I think some of these factors should be impacted, therefore, please more discuss the result and give some reasons for that. “pandemic effect on daily life” should be impacted, Perhaps these results are due to the high percentage of people who want to get vaccinated.

-          What can be the connection between marriage and not wanting to be vaccinated?

In some places the willingness for vaccination was very low, Karnali province OR Sudurpaschim, why? Any logical reason? One of my concerns is the choice of respondents.

For better results, I suggest that select a cutoff for these factors (For better results I suggest that/ Pandemic impact on work/ Pandemic impact on income) and categorize them into two items.

Best

Round 2

Reviewer 1 Report

The presentation of study details has been improved.

Reviewer 2 Report

All other questions have been taken in consideration, this paper can be considered for publication .